



5 # An improved post-processing technique for automatic precipitation gauge time series

Amber Ross[1,2], Craig D. Smith[1], and Alan Barr[1,2]

10 [1]Environment and Climate Change Canada, Climate Research Division, Saskatoon, SK
[2]Global Institute for Water Security, University of Saskatchewan, Saskatoon, SK

*Correspondence to:* Craig D. Smith (craig.smith2@canada.ca)

15 **Abstract.** The unconditioned data retrieved from automated accumulating precipitation gauges is inherently noisy due to the sensitivity of the instruments to mechanical and electrical interference. This noise, combined with diurnal oscillations and signal drift from evaporation of the bucket contents, can make accurate precipitation estimates challenging. Relative to rainfall, errors in the measurement of solid precipitation are exacerbated because the lower accumulation rates are more impacted by measurement noise. Precipitation gauge measurement post-processing techniques are used by Environment and Climate Change Canada in research and operational monitoring to filter cumulative precipitation time series derived from high-frequency, bucket-weight measurements. Four techniques are described and tested here: 1) the operational 15-minute filter (O15), 2) the Neutral Aggregating Filter (NAF), 3) the Supervised Neutral Aggregating Filter (NAF-S), and 4) the Segmented Neutral Aggregating Filter (NAF-SEG). Inherent biases and errors in the first two post-processing techniques have revealed the need for a robust automated 25 method to derive an accurate noise-free precipitation time series from the raw bucket-weight measurements. The method must be capable of removing random noise, diurnal oscillations, and evaporative (negative) drift from the raw data. This evaluation focuses on cold-season (October to April) accumulating-precipitation-gauge data at 1-min resolution from two sources: a control (pre-processed time series) with added synthetic noise and drift; and raw (minimally-processed) data from several WMO Solid Precipitation Inter-Comparison Experiment (SPICE) sites. 30 Evaluation against the control with synthetic noise shows the effectiveness of the NAF-SEG technique, recovering 99%, 100%, and 102% of the control total precipitation for low, medium, and high noise scenarios respectively. Among the filters, the fully-automated NAF-SEG produced the highest correlation coefficients and lowest RMSE for all synthetic noise levels, with comparable performance to the supervised and manually-intensive NAF-S method. Compared to the operational O15 method, NAF-SEG shows a lower bias in 37 of 44 real-world test cases, a similar 35 bias in 5 cases, and a higher bias in 2 cases. The results indicate that the NAF-SEG post-processing technique provides substantial improvement over current automated techniques, reducing both uncertainty and bias in accumulating-gauge measurements of precipitation, with a 24-hour latency. Because it cannot be implemented in real time, we recommend that NAF-SEG be used in consort with a simple real-time filter, such as the operational O15 or similar filter.

40



## 1 Introduction

Accurate precipitation measurements are crucial for a variety of applications, including water resource forecasting, future water availability, and hydrological and climate analysis and modelling (Barnett et al., 2005; Bartlett et al., 2006; Wolff et al., 2015). Canada's Changing Climate Report led by Environment and Climate Change Canada (2019) highlights the importance of accurate precipitation measurements as fundamental climate quantities that play an important role in human and natural systems. Although the systematic bias due to the impact of wind on solid precipitation measurements is well documented (Goodison, 1978; Sevruk et al., 1991; Goodison et al., 1998;Yang et al 2005; Sevruk et al., 2009; Smith, 2009; Wolff et al., 2015; Kochendorfer et al., 2017a), errors related to the automatic recording of precipitation measurements have only relatively recently been identified as automated weighing gauges come into common use (Sevruk, 2005). The cumulative precipitation data output from automated weighing gauges is subject to noise, diurnal temperature oscillations, and negative drift from evaporation which can often mean that the precipitation signal over short sampling periods is influenced or hard to detect (Rasmussen et al., 2012). The nature of the noise and drift often varies substantially from site to site and between gauge configurations. High frequency noise can exceed ± 1 mm and evaporation from the bucket can be in excess of several mm between precipitation events. It is therefore necessary to filter the raw data to separate real precipitation events from signal noise and identify and remove periods with evaporation (keeping in mind that evaporation reduces the precipitation amount derived from the differential in bucket weight). Improper filtering can lead to the accumulation of errors and result in significant inaccuracies in total seasonal precipitation. Duchon (2008) suggests that errors due to the diurnal oscillation in Geonor T-200B gauges could be 1-10% of the precipitation total. Three post-processing challenges in the derivation of 'clean' precipitation time series are the focus of this study: mechanical and electrical interference, diurnal oscillations, and evaporation of the bucket contents.

This study incorporates two commonly-used automated weighing gauges: the Geonor T-200B and OTT Pluvio[2]. The Geonor T-200B implements up to three vibrating wire transducers, which provide a frequency output that varies as a function of the fluid weight in the gauge bucket. The cumulative precipitation amount (bucket weight) is calculated from the frequency of each wire via calibration coefficients, with no onboard filtering (Geonor, 2019). The OTT Pluvio[2] precipitation gauge uses a high-precision load cell to weigh the bucket contents and provides several outputs including intensity and precipitation accumulation (Nemeth, 2008; Nitu et al., 2018). The OTT Pluvio[2] output has been pre-processed using an onboard proprietary algorithm which adjusts the high frequency load cell measurements for temperature and vibration to derive a more accurate bucket weight. Further onboard processing removes the impact of unrealistic bucket weight changes and evaporation from the output, however, some would prefer to bypass the onboard algorithm and complete their own post-processing of the data in its rawest form.

A number of post-processing techniques have been developed to derive a noise-free precipitation time series from high-frequency gauge bucket weight measurements. Some examples are described here.

The Rolling Maximum filter was used by Harder and Pomeroy (2013) to remove the "jitter" from the accumulated precipitation datasets by retaining a cumulative precipitation observation if it is greater than the previous maximum



cumulative precipitation. The previous maximum is assumed to be the cumulative precipitation in all other cases. This filter reportedly works well in preserving the cumulative change in precipitation but it may not always catch the precise start of precipitation events and will not always perform optimally in the presence of negative gauge drift (i.e.

evaporation).

The World Meteorological Organization (WMO) Solid Precipitation Inter-Comparison Experiment (SPICE, 2013-2015) developed a uniform post-processing method for defining and quantifying precipitation events (Nitu et al., 2018). The process includes calculating a 30-minute bucket weight differential using thresholds and filters, effectively producing what was termed the Site Event Datasets (SEDS). For an event to be identified, the net precipitation duration

needed to be sufficiently long (as measured by a precipitation-detector or disdrometer) and the total accumulation (as measured by the reference weighing gauge) needed to be equal to or greater than a defined threshold (set at 0.25 mm when a reliable precipitation-detector was available). This process was effective at creating a high confidence data set for developing and testing transfer functions (Kochendorfer et al., 2017b) but because of the rigorous filtering of shorter and smaller events, was not an effective means of filtering a time series.

The U.S. Climate Reference Network (USCRN) uses the redundancy of the Geonor T-200B three vibrating-wire load sensors in the determination of precipitation events (Leeper et al., 2015). Initially, a pairwise calculation was used which relies on pairwise agreement of bucket weight changes using the wire redundancy as a check on the measurement. This was determined to be sensitive to gauge evaporation and noise, leading to the development of a weighted average calculation using the change in bucket weight between successive sub-hourly periods for each

transducer output. A weighted mean is then used to average the bucket weights, with greater weight given to less noisy measurements.

The Meteorological Service of Canada currently implements a real-time threshold filter in their data loggers to automatically determine the occurrence of precipitation events. The filter is based on the 15-min differential in the Geonor T-200B bucket weight (Mekis et al., 2018). Although this filter is unnamed, we call it the Operational 15

Minute (O15) automated processing technique. This technique is included in this analysis and is described below in more detail. The filter tends to fail when the noise threshold is exceeded, resulting in false precipitation reports, and when evaporation exceeds the acceptable limits.

Limitations in the O15 technique led to the development of the Neutral Aggregating Filter (NAF), previously known as 'Brute Force' (Pan et al., 2016). The NAF, described in greater detail by Smith et al. (2019), iteratively adds all

negative and small positive changes to proximate positive changes until all changes exceed a user-specified threshold. Because the technique preserves the total change in bucket weight over the time series, it cannot account for the negative drift that results from evaporation. To overcome this deficiency, the Supervised Neutral Aggregating Filter (NAF-S) was created to allow user intervention and minimize evaporation errors through interactive manual adjustment. Both NAF and NAF-S are explained in greater detail in the next section.

To overcome the limitations of the O15, NAF, and NAF-S techniques, we evaluated a moving-window modification of the NAF, implementing the NAF on 24-hour overlapping windows, which we will call the Segmented Neutral



Aggregating Filter (NAF-SEG). The objective was to obtain a robust post-processing technique that is completely automated, easily implemented, and successfully eliminates varying levels of noise, diurnal oscillations and evaporation without significantly impacting the timing and amount of precipitation. This study introduces the NAF-

SEG technique and examines its performance compared to the O15, NAF, and NAF-S methods.

## 2 Processing Techniques Under Test

### 2.1 MSC Operational O15 Minute

The O15 filtering technique is used operationally by the Meteorological Service of Canada (MSC) for Geonor T-200B

measurements at the Reference Climate Stations (RCS). The O15 filter is implemented in real time at the measurement site data logger. The algorithm is intended to filter out noise and eliminate evaporation while minimizing the reports of false precipitation. For each 15-min period, a mean bucket weight is computed over the last 5 min (minutes 11 to 15) of the period. The mean bucket weight from the initial period is used to establish the baseline. For each successive 15-min period, the difference between the current mean bucket weight and the baseline is calculated. If the bucket

weight difference is greater than or equal to 0.2 mm, the difference is attributed to precipitation and added to the cumulative precipitation total, and the baseline is reset upwards to the current mean. If the difference is less than or equal to -1.0 mm, the difference is attributed to evaporation and the baseline is adjusted downward to match the current mean. This process is performed separately on each of the three installed transducers in the RCS gauge although ultimately only one is used to determine reported precipitation.

The O15 technique is used operationally in real-time, and so must be simpler than other post-processing techniques. As a result, it has the potential to be problematic, including a sensitivity to the positive and negative thresholds used to identify precipitation and evaporation events. The 0.2 mm positive accumulating (noise) threshold can cause an overestimation of precipitation if the data are inherently noisy or have a high diurnal oscillation. Additionally, if the negative drift from evaporation lies just above the -1.0 mm threshold, the baseline will not be adjusted before the next

precipitation event, resulting in an underestimation of the next event by up to 1.2 mm (evaporation threshold plus the noise threshold).

### 2.2 Neutral Aggregating Filter

The NAF method, developed by Environment and Climate Change Canada's Climate Research Division, is an automated method that removes noise from cumulative precipitation time series (Pan et al., 2016; Smith et al., 2019).

The processing is done iteratively, beginning with the minimum non-zero interval precipitation value. All non-zero changes in interval precipitation, with values below a user-defined threshold are transferred to neighboring periods with positive or larger changes. The results from the algorithm are "neutral" as the filter balances the positive and negative noise until all changes below the user-defined threshold are eliminated.



The technique removes random noise and accounts for diurnal oscillations in the bucket-weight signal but, because the total precipitation is forced to equal the total bucket weight increase at the end of the time series, it cannot account for negative drift. This means that it will not perform well if the time series has significant periods with evaporative losses from the accumulating gauge bucket. The significance of the error could exceed 10% depending on the effectiveness of the servicing measures to reduce evaporation from the bucket contents. NAF serves as the framework for both the NAF-S and NAF-SEG techniques described below.

In this study, the NAF, NAF-S (2.3) and NAF-SEG (2.4) methods all use a minimum threshold $P*$ of 0.001 mm. $P*$ was somewhat arbitrarily set at 0.001 mm based on the minimum resolution of the gauge data. Testing (not shown here) suggests that the method is not overly sensitive to $P*$ and that a 5-fold increase in the magnitude of $P*$ had minimal impact on the performance.

### 2.3 Supervised Neutral Aggregating Filter

The NAF-S method is used to manually adjust the cumulative time series for evaporation and other spurious data, effectively reducing the NAF estimation error. The NAF-S method uses the NAF output as a first guess, and then allows for manual, interactive adjustment of the baseline to account for evaporation events and other data artifacts impacting the time series. The NAF-S creates an interactive plot, showing both raw (quality controlled) and NAF output data, which highlights periods with drift caused by evaporation. The user is then given the capability to identify and manually exclude each period with evaporation, using the cumulative precipitation value before each evaporation event as a new baseline. NAF-S successfully minimizes the impact of evaporation but requires user intervention (i.e. it cannot be automated) along with user subjectivity to identify the endpoints of evaporative and other spurious events (Smith et al., 2019).

### 2.4 Segmented Neutral Aggregating Filter

The NAF-SEG is a fully automated technique that implements the NAF to process multi-day precipitation time series in successive 24-hour segments using overlapping moving windows. The use of 24-hour windows automates the identification and removal of evaporation, minimizing the negative biases in total precipitation from evaporation without the need for user intervention. Additionally, the NAF-SEG method provides an estimate of evaporative losses on precipitation-free days for evaluating servicing procedures.

The NAF-SEG technique uses three overlapping moving windows per day, advanced in increments of 8 hours. The algorithm begins by filtering the first 24-hour segment using NAF. It then advances 8 hours and filters the next 24-hour segment. This filtering process is repeated until the end of the data is reached. Each 8-hour data segment thus passes through the NAF three times. The processing steps are listed below and outlined in Fig. 1.

We will denote the precipitation amount from one measurement interval (i) as P(i); cumulative precipitation as cumP(i); evaporation from one measurement interval as E(i) ; and cumulative evaporation as cumE(i). All units are in mm.




1. The time series is processed in successive 24-hour segments.

2. For each 24-hour segment, the change in bucket weight, which we will call $\Delta^{24h}$, is computed as the difference between the final and initial observations.

3. Based on the value of $\Delta^{24h}$, the 24-hour segment is assigned one of three states: 1) precipitating, 2) evaporating, or 3) neither. It is then processed accordingly:

   a. If $\Delta^{24h} \geq P^*$, the 24-hour segment is flagged and treated as a precipitation period with no evaporation. The 24-hour segment is passed through the NAF, resulting in values of $P(i)$ that are either zero or greater than or equal to $P^*$.

b. If $\Delta^{24h} \leq -P^*$, the 24-hour segment is flagged and treated as an evaporation period with no precipitation. The 24-hour segment is passed through the NAF but with the sign of the data reversed, resulting in values of $E(i)$ that are either zero or less than or equal to $-P^*$.

   c. If $-P^* < \Delta^{24h} < P^*$, the 24-hour segment is flagged as free of both precipitation and evaporation, and all values of $P(i)$ and $E(i)$ are set to zero.

4. The NAF $P(i)$ and $E(i)$ outputs from step (3), as well as the flags that indicate the presence of precipitation or evaporation, are added to arrays with three columns corresponding to the three overlapping windows per day (i.e. as $P(i,j)$, $E(i,j)$ and $flag(i,j)$ where $j$ denotes columns (windows) 1 to 3).

5. Steps (2) to (4) are repeated using moving windows on successive 24-hour segments, beginning 8 hours apart, until the entire time series has been processed.

6. The $P(i,j)$ and $E(i,j)$ arrays from steps (3) to (5), with three overlapping windows, are processed to create a single time series for $P(i)$ and $E(i)$, based on the *flag*.

   a. For intervals when the *flag* from all three overlapping windows indicates the presence of precipitation, $E(i)$ is set to zero and the three $P(i,j)$ values are averaged to produce $P(i)$, otherwise $P(i)$ is set to zero.

   b. For intervals when the *flag* from all three overlapping windows indicates the presence of evaporation, $P(i)$ is
set to zero and the three $E(i,j)$ values are averaged across columns to produce $E(i)$, otherwise $E(i)$ is set to zero.

   c. For intervals when the *flag* from all three overlapping windows does NOT indicate the presence of precipitation or evaporation, $P(i)$ and $E(i)$ is set to zero.

7. The $P(i)$ and $E(i)$ outputs from step (6) are summed to create the *cumP* and *cumE* time series. Lastly, *cumP* is
passed through the NAF to ensure that all $P(i)$ values are either zero or greater than or equal to $P^*$; *cumE* is passed through the NAF but with the sign of the data reversed to ensure that all $E(i)$ values are either zero or less than or equal to $-P^*$. The evaporation estimate is taken as the absolute value of the cumulative total of *cumE*.

Two additional steps not shown in Fig. 1 are required. First, additional 24-hour segments need to be added to the start and end of the time series to ensure that all core intervals are covered by three overlapping windows. Since these time
series begin at 0 mm at the start of the season, the 24-hour segment added to the start of each time series is set to all zero values. The 24-hour segment added to the end of the time series is set to the maximum of the cumulative time series. This step is only necessary if the user requires processed data from the first and last 24-hour period in the time series and does not impact the precipitation amounts.





A second step is required to ensure that the precipitation during data gaps is not omitted from the accumulated total.
Note that when gaps occur in a weighing gauge time series, the total accumulation across the gap is preserved but the
event timing is lost. In the NAF-SEG implementation, precipitation occurring over data gaps is preserved if all three
windows capture the jump in the bucket weight over the gap. But this will not always be the case. We resolved the
problem as follows. First, we identified data gaps that overlapped the start or end of each 24-hour segment, computed
the difference in bucket weight across the gap, and flagged windows when the difference was greater than or equal to
$P*$. For those segments only, we added a processing step between steps (5) and (6), as follows. If any of the three
overlapping windows captured the jump in the bucket weight across the gap, the window(s) in P(i,j) that did not
capture the jump were excluded from the averaging, and all three windows were flagged to indicate the presence of
precipitation. If none of the windows captured the jump in bucket weight across the gap, the difference across the gap
was assigned to the final interval of the gap in P(i,j) for all three windows, with all windows flagged to indicate the
presence of precipitation.

## 3  Filter Evaluation

Two data sources, both with 1-min resolution, were used to evaluate the O15, NAF, NAF-S and NAF-SEG
precipitation filters: a control (pre-processed) precipitation time series which is free of noise and drift; and raw
(minimally filtered) accumulating gauge data collected at a number of international sites, which contain varying
levels of noise, diurnal oscillations, and evaporative drift. The clean, pre-processed time series were used to evaluate
all four filters -- by adding synthetic noise, diurnal oscillations, and evaporative drift, then evaluating the ability of
the filters to recover the original time series. The raw time series were passed through each of the filters, and the
supervised NAF-S output was used as the standard against which to evaluate the others.

Both data sources, raw data with real-world noise, and clean data with synthetic noise added, have advantages and
disadvantages in assessing filter performance (Peters et al., 2014). Clean data with added noise provide a known
'true' control but add the risk that the added noise and drift may not adequately capture the characteristics of real-
world measurements. Raw measurements preserve observed noise patterns and capture the variability in noise
behavior across sites and instruments, but do not provide a control time series for filter evaluation. By using both
complementary data sources, we exploit their respective strengths and thus better assess the relative effectiveness of
each filter.

### 3.1 Testing with pre-processed (control) precipitation data

The pre-processed 1-minute cumulative time series was originally derived from an Alter-shielded Geonor T-200B
precipitation gauge at Caribou Creek, Canada from October-2013 through April-2014. The raw gauge outputs were
filtered using NAF-S, resulting in a precipitation total of 259 mm. Historically, this particular gauge has performed
well with minimal noise ($< \pm 0.25$ mm) and evaporation issues; the time series was very clean even prior to filtering,
and therefore the filtered output provides a suitable control.





To evaluate the four filters, we added synthetic noise and drift to the filtered (noise-free) control, then tested each filters ability to recover the original signal. The perturbations included synthetic evaporation, diurnal oscillations, and
random noise, computed as follows:

1.  Negative evaporative drift was added that totaled 25.9 mm or 10% of the precipitation total. The synthetic evaporation was partitioned among the 1-min intervals assuming that interval evaporation was proportional to the vapor pressure deficit (VPD). The fraction of evaporation for each interval was calculated by dividing the interval VPD by the VPD sum over the entire time series. Those fractions were then multiplied by the total (25.9 mm),
and the resulting cumulative sum was subtracted from the control cumulative precipitation.

2.  Temperature-dependent diurnal oscillations $\delta^T(i)$ were computed from observed air temperature at gauge height and added to the cumulative precipitation control. The diurnal oscillations were calculated as:

$$\delta^T(i) = fTs * (T(i) - mean(T)) / (0.5*range(T) \qquad (1)$$


where fTs is a coefficient that varies for the different noise scenarios (Table 1). The temperature-oscillation time series $\delta^T$ was then subtracted from the cumulative time series from Step 1.

3.  Normally-distributed random noise was generated for each 1-min interval, with a mean of zero and a specified standard deviation (Table 1). Because the synthetic noise time series is generated randomly, it does not necessarily
sum to zero. To avoid adding bias, we forced the sum to zero by subtracting the mean. The result was then added to the cumulative time series from Step 2.

The artificially-noisy time series from Step 3 was adjusted to a value of zero at the start, and then filtered using the O15, NAF, NAF-S, and NAF-SEG techniques.

### 3.2  Testing with raw precipitation data

Precipitation-gauge data were collected between 2013 and 2017 at seven WMO-SPICE (Nitu et al., 2018) sites including Bratt's Lake (XBK; Canada), Caribou Creek (CCR; Canada), Centre for Atmospheric Research and Experiments (CAR; Canada), Formigal (FMG; Spain), Haukeliseter (HKL; Norway), Sodankylä (SOD; Finland), and Weissfluhjoch (WFJ; Switzerland). These sites provided high-quality precipitation observations from several automated precipitation gauge (Geonor T-200B and OTT Pluvio²) configurations at a temporal resolution of 1-minute.
In addition, the sites utilized a number of wind-shield configurations including the WMO Double Fence Automated Reference (DFAR), and the single Alter-shield, as well as unshielded configurations. The combination of different climate regimes, gauge types, and wind-shield configurations, provides the opportunity to test processing algorithms on contrasting noise patterns. Although the SPICE intercomparison period (2013-2015) officially ended in 2015, many of these high-quality precipitation observations were continued beyond 2015 and made available by the site hosts for
this evaluation.

In total, 44 winter time series (from October through April over years 2013 to 2017) were used in testing. The raw 1-minute data were first run through an automated quality control process to remove out-of-range outliers and data



jumps, which included the removal of data jumps/drops related to gauge servicing (bucket emptying and/or charging) consistent with the quality control process used for the WMO-SPICE analysis (Nitu et al., 2018). Anything missed or flagged by the automated quality control process was examined and, as necessary, cleaned manually. The 1-minute precipitation bucket-weight data were then smoothed using a Gaussian filter with a 4-minute running window. This filter smoothed large spikes in the time series that may have resulted from mechanical or electrical noise. Since all of the Geonor T-200B gauges used in this analysis were equipped with three vibrating wire transducers, the bucket weights from each wire were averaged following the quality control process to derive a single time series. This has been shown to further reduce random noise (Duchon, 2008). Finally, the time series were zeroed at the start of the season and the cumulative time series was filtered using the O15, NAF, NAF-S, and NAF-SEG techniques.

Unlike the first data sources, the raw (minimally-filtered) observations do not provide a control. To overcome this limitation, we used the NAF-S output as the reference standard for the other three methods. This adds a potential bias because of NAF-S-user subjectivity, but we believe the bias to be small. Previous tests have shown NAF-S to achieve favorable results (Smith et al., 2019).

### 3.3 Analysis methods

For analysis, the 1-minute filtered data were aggregated into 30-minute accumulation intervals. Three statistical tests were chosen to analyze the performance of the post-processing techniques: total bias (for each seasonal time series), root mean square error (RMSE; or more appropriately, root mean square deviation RMSD for the tests with unfiltered data), and Pearson's correlation coefficient ($r$). The total bias is a valuable metric that demonstrates the post-processing technique's overall ability to generate an accurate total. The RMSE (or RMSD) quantifies the variability of the filter outputs relative to the control or reference standard. Finally, Pearson's correlation coefficient determines the strength of the linear relationships between the filter outputs and the control or reference. RMSE (or RMSD) and $r$ are based on the interval precipitation amounts and include the intervals with zero precipitation.

## 4 Results

### 4.1 Filter evaluation using pre-processed (control) data

The performance of the four filters was evaluated by adding synthetic noise and drift to a clean (control) time series and then assessing each filter's skill in recovering the control. The results are shown in Fig. 2, and an in-depth look at the first simulated evaporation event is shown in Fig. 3, for each of the three noise scenarios. Tables 2 to 4 show the associated 30-minute total seasonal biases, correlation coefficients, and RMSE for all four filters, and the NAF-SEG evaporation estimates.



Based on their success in eliminating the added synthetic noise and drift and recovering the original control time series, NAF-S and NAF-SEG outperformed NAF and O15. O15 performed well at low noise but was sensitive to higher noise levels, with biases in total precipitation of +1%, +13% and +33% for the low, medium and high noise scenarios respectively. NAF was insensitive to noise but failed to recover the added evaporative losses (10% of the precipitation total) at all noise levels. NAF-S and NAF-SEG performed well at all three noise levels, recovering the

control precipitation to within 2% of the total and generating the highest correlation coefficients and lowest RMSE. NAF-SEG also produced an estimate of evaporation; its skill in detecting evaporative losses varied by noise level, with a 16% overestimation of the synthetic total at high noise and a 19% underestimation at low noise. Given the inherent difficulty of the task and the high degree of temporal detail in the added evaporation time series, and given that the fully automated NAF-SEG matched the skill of the manually supervised NAF-S, the ability of the NAF-SEG

filter to detect and eliminate negative drift was encouraging.

### 4.2 Filter evaluation using unprocessed data

This intercomparison examines the relative performance of the O15, NAF and NAF-SEG filters on raw (minimally-processed) weighing-gauge time series, using the NAF-S output as the reference standard. Individual results from the 44 test time series are shown in Table A1. Overall, the NAF-SEG technique gave the lowest mean bias, highest mean

correlation coefficient r, and lowest mean RMSD value (Table 5). The absolute bias from NAF-SEG was lower than the O15 bias in 37 of 44 cases (84%), similar in 5 cases (11%) and higher in 2 cases (5%). NAF-SEG also produced the lowest variability in r, RMSD and seasonal total (Fig. 4), suggesting the greatest consistency in processing performance across sites, configurations and years.

The relative performance of NAF-SEG, NAF, and O15 varied across the 44 test time series, related to the nature and

magnitude of the noise and negative drift due to evaporation from the bucket (Table A1). Figure 5 shows four examples, comparing raw and processed time series. The y-axis is scaled to the precipitation total to provide perspective on the relative errors in the processing techniques. The inset graphs in Fig. 5, which zoom in on particular events, highlight the magnitude of noise and drift in the raw data and show how the filters respond.

Figure 5a shows a time series for Caribou Creek (CCR), Canada, where the raw data exhibits very little noise or

evaporation. For that reason, all processing techniques are within a few percent of the NAF-S reference, and it is difficult to see the differences during much of the time series. Fig. 5b, from Haukeliseter (HKL), Norway, exhibits higher noise, resulting in an O15 precipitation overestimate of +9% due to false precipitation detection. A moderate amount of evaporation is seen in the growing difference between NAF and NAF-S, with NAF-SEG nearly replicating NAF-S. Fig. 5c and 5d, from Bratt's Lake (XBK), Canada, show cases with high evaporation (5c) and high noise (5d).

In Fig. 5c, evaporation causes a low bias in NAF, which recovers only 87% of the NAF-S precipitation total; O15 shows two compensating errors – an underestimation in precipitation due to evaporation and an increase in false precipitation detections due to noise, resulting in a recovery of 94% of total precipitation relative to NAF-S; and NAF-SEG closely replicates NAF-S, with slight deviations in Nov. and Dec. Fig 5d shows the impact of high noise with



little evaporation; O15 overestimates precipitation by 4%, whereas NAF-SEG is consistent with NAF-S throughout
the time series.

## 5  Discussion

This study evaluated four filters for processing the outputs of accumulating (weighing) precipitation gauges, three
that were fully automated (O15, NAF and NAF-SEG) and one that required manual supervision (NAF-S). Overall,
NAF-S and NAF-SEG outperformed O15 and NAF; both NAF-S and NAF-SEG showed similar skill in
compensating for evaporative losses and eliminating false detections caused by random noise and diurnal
oscillations. O15 performed well in low noise cases with minimal evaporation, but generated false precipitation
detections when the data were noisy, and often underestimated evaporative losses. NAF performed well in cases
with minimal evaporation regardless of the noise level but did not correct for evaporative losses. NAF-SEG
performed consistently well and provided a fully-automated alternative that matched the skill of the manual NAF-S
method. Moreover, NAF-SEG added a direct estimate of evaporation, without the user intervention required by
NAF-S or the 1-mm threshold required by O15. Similar evaporation estimates are not directly available from the
other techniques.

Although NAF-SEG did not perfectly recover the synthetic evaporation that was added to the control time series (the
recovery rates were 81% to 116% depending on the noise level), it performed as well as the manually-supervised
NAF-S technique. Both NAF-S and NAF-SEG failed to disentangle precipitation and evaporation when they
occurred on the same day. The challenge to do so may be insurmountable. The imperfect recovery of synthetic
evaporation, coupled with the sensitivity of the recovered evaporation to noise, highlights the need to implement
measurement protocols that minimize evaporative losses. We recommend the use of NAF-SEG as a screening
technique to identify gauges and locations that have significant evaporative losses, and then to implement adequate
measures to minimize those losses, such as modifications to the oil and antifreeze mixture used to prevent freezing
and evaporation.

Overestimation of precipitation by the O15 method occurs when the noise exceeds the filter's prescribed threshold
of 0.2 mm. This value for the threshold has been set based on experience as a necessary and calculated balance
between eliminating real precipitation events and detecting false events. When the noise level is low, as in the low
noise scenario of the control data, the O15 technique works successfully. However, noise patterns vary substantially
from site to site and among gauges, as illustrated by Nitu et al. (2018), and often exceed the filtering capabilities of
O15.

The NAF technique is fundamentally effective at filtering noise and diurnal oscillations, but underestimates
precipitation when evaporative losses occur, because the algorithm forces precipitation total to match the final raw
bucket weight in the time series, with evaporation assumed to be zero. The NAF-SEG technique, which implements
NAF over 24-hour windows, maintains all the strengths of NAF with the added functionality of automating the


detection and removal of bucket evaporation. Neither NAF-S or NAF-SEG remove evaporation perfectly, particularly when it occurs in consort with precipitation, but both represent a major step forward compared to other

processing methods. We attribute the effectiveness of NAF-SEG to two properties of precipitation events, first that evaporation is relatively small during periods with precipitation, and second that both precipitation and evaporation are persistent over time scales of days. In the development of NAF-SEG, a 24-hour moving window was chosen to minimize the impact of temperature-related diurnal oscillations, but fortuitously the 24-hour window also serves to separate days with precipitation and little evaporation from days with evaporation and little or no precipitation.

NAF-SEG provides an attractive alternative to NAF when negative evaporative drift is present in the raw data, but it is not designed to handle all contingencies. For instance, unexplained positive then negative excursions in bucket weight are sometimes observed. If the positive and negative excursions are separated by more than 24 hours (the size of the window), the NAF-SEG filter will errantly attribute the positive excursion to precipitation and the negative excursion to evaporation.

Of the 44 raw-data test cases from different sites, seasons, and gauge configurations, O15 outperformed NAF-SEG in only two cases: the DFAR and unshielded Pluvio$^2$ gauges at WFJ, 2016-2017. These gauges may not have been serviced adequately; note the extreme evaporation rates as evidenced in the high biases between NAF and NAF-S in Table A1. This may diminish the value of these time series for this evaluation; they were among the most challenging to process, adding uncertainty to the supervised NAF-S output which served as the reference standard.

This observation begs the question: how reliable are the NAF-S outputs as reference standards, given that they rely on the operator's subjective judgment during the interactive elimination of negative drift and other spurious bucket weight changes? We acknowledge that operator bias is possible but are confident that its impact in this study is minimal. A single, skilled operator processed all of the data and made every attempt to apply the NAF-S method consistently. Adding further confidence to the NAF-S outputs are the tests with control data, which independently

demonstrated the efficacy of the NAF-S to eliminate noise and evaporative drift.

The precipitation time series used in his study were collected during October to April as part of an intercomparison of solid-precipitation measurement techniques. Although the data include liquid, mixed, and solid precipitation events at many of the sites, the precipitation is predominately snowfall. We recommend a complementary follow-up study on the processing of warm-season precipitation measurements from accumulating gauges. The issues faced

may vary by season. Compared to other seasons, cold-season evaporation rates are low, as was confirmed for most cases in this study (Table A1). The identification and elimination of evaporative losses may be more challenging in the warm season when evaporation rates are higher. On the other hand, precipitation rates are generally lower for solid precipitation than rainfall, which reduces the signal to noise ratio, therefore adding difficulty to the processing of solid precipitation data.




## 6 Conclusions

This study reports the development and implementation of a robust, fully-automated technique for post-processing data from accumulating (weighing) precipitation gauges. The NAF-SEG technique is designed to eliminate varying

levels of random noise and diurnal oscillations, as well as correcting for negative drift from bucket evaporation. An intercomparison of four filtering techniques shows that the operational O15 filter, although simple and deployable in real-time, fails when noise levels exceed the filter's threshold, and may undercompensate for bucket evaporation. NAF, although highly effective in eliminating noise, does not correct for evaporative losses. NAF-S, which adds manual supervision to NAF, is effective in removing noise, eliminating spurious data, and correcting for negative

drift from evaporation. However, it is labour intensive and best suited to complete seasonal time series.

Our results show that NAF-SEG is equally effective to NAF-S in eliminating noise and evaporative drift from accumulating-gauge precipitation measurements. When tested against a control data set with added synthetic noise and evaporation, NAF-SEG was able to recover the original control to within ± 2% of the total, with a lower RMSE than the other techniques. When evaluated on 44 raw time series from various sites, years and gauge configurations,

NAF-SEG outperformed O15 and NAF and gave the highest mean correlation coefficient and lowest mean RMSD.

One limitation of NAF-SEG that it requires 24-hour data segments; consequently, it cannot be deployed for real-time processing of weighing-gauge precipitation measurements. Until other alternatives are found, we recommend the use of simple threshold filter like O15 for real-time applications, but with the archiving of the raw 1-min time series for subsequent reprocessing using NAF-SEG, and the archiving of the NAF-SEG outputs.


### Acknowledgments

The authors would like to thank Dr. Daqing Yang of Environment and Climate Change Canada (Victoria) for providing an internal review of this manuscript. We would like to acknowledge the organizations that collected and provided the observed data from the WMO-SPICE sites for this analysis: Environment and Climate Change Canada

(Climate Research Division and the Meteorological Service of Canada Observing Systems and Engineering), the Norwegian Meteorological Institute, MeteoSwiss, the Finnish Meteorological Institute, and the Spanish National Meteorological Agency.

### Data and code availability

The code for NAF-SEG and the precipitation time series intercomparison data used in this evaluation will be made available in a suitable online repository.



**Author contribution**

A.R. is the lead author and was responsible for the processing and analysis of these data. A.R. also completed much of the coding required to implement the data processing. C.S. oversaw the development of this project and provided guidance in the analysis and the development of this manuscript. A.B. designed and coded the NAF filters and provided guidance in the analysis and in the writing of this manuscript.

**Competing interests**

The authors declare that they have no conflict of interest.

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





**Table 1: Diurnal and random noise parameters in the simulated precipitation time series**

| Noise Level | High | Medium | Low |
|---|---|---|---|
| Diurnal coefficient (fTs) (mm) | 2 | 1.5 | 1 |
| Random noise (std dev) (mm) | 0.1 | 0.01 | 0.001 |

**Table 2: Total seasonal bias in mm and percent of total for NAF, NAF-S, O15, and NAF-SEG post-processing techniques at different simulated noise levels.**

| Noise Level | NAF (mm) | NAF (%) | NAF-S (mm) | NAF-S (%) | O15 (mm) | O15 (%) | NAF-SEG (mm) | NAF-SEG (%) |
|---|---|---|---|---|---|---|---|---|
| Low | -26.1 | -10.1% | -6.4 | -2.5% | 1.5 | +0.6% | -2.8 | -1.1% |
| Med | -26.2 | -10.1% | -3.4 | -1.3% | 33.5 | +12.9% | 1.0 | +0.4% |
| High | -26.3 | -10.2% | -2.7 | -1.0% | 86.0 | +33.2% | 5.0 | +1.9% |

**Table 3: Correlation coefficient (r) and RMSE for NAF-SEG, NAF-S, NAF, and O15 post-processing techniques at different simulated noise levels.**

| Noise Level | r NAF | r NAF-S | R O15 | r NAF-SEG | RMSE NAF (mm) | RMSE NAF-S (mm) | RMSE O15 (mm) | RMSE NAF-SEG (mm) |
|---|---|---|---|---|---|---|---|---|
| Low | 0.97 | 0.99 | 0.94 | 0.99 | 0.029 | 0.020 | 0.044 | 0.019 |
| Med | 0.97 | 0.98 | 0.92 | 0.98 | 0.032 | 0.025 | 0.053 | 0.024 |
| High | 0.95 | 0.96 | 0.87 | 0.96 | 0.041 | 0.038 | 0.069 | 0.037 |


**Table 4: NAF-SEG evaporation estimates for different simulated noise levels with actual evaporation constant at 25.9 mm in the control.**

| Noise Level | Recovered Evaporation (mm) | % of Actual |
|---|---|---|
| Low | 21.0 | 81% |
| Med | 25.1 | 97% |
| High | 30.1 | 116% |





**Table 5: Mean correlation coefficients (r) and RMSD along with standard deviations (SD) for all observed real-world precipitation time series using NAF-S as the reference.**

| Post-Processing Technique | Mean r | SD r | Mean RMSD (mm) | SD RMSD (mm) |
|---|---|---|---|---|
| NAF-SEG | 0.99 | 0.006 | 0.017 | 0.006 |
| NAF | 0.98 | 0.040 | 0.020 | 0.025 |
| O15 | 0.95 | 0.032 | 0.041 | 0.024 |





**Appendix A: Raw time series used in precipitation filter evaluation, with evaporation estimates and total**

**precipitation bias.**

**Table A1:** Seasonal total precipitation (unfiltered, NAF-S and NAF-SEG filtered), filter biases (NAF, O15 and NAF-SEG) and derived bucket evaporation (NAF-SEG) from 44 WMO-SPICE precipitation time series. Biases (mm) are calculated using NAF-S as the reference filtering technique. Filtered time series that do not show an improvement with the NAF-SEG method when compared to O15 are indicated by an asterisk (*).

| Site/Shield/Gauge/Year | Unfiltered Total (mm) | NAF-S Total (mm) | NAF-SEG Total (mm) | Bias NAF (mm) | Bias O15 (mm) | Bias NAF-SEG (mm) | Evaporation NAF-SEG Estimate (mm) |
|---|---|---|---|---|---|---|---|
| CAR-R2P-2016-2017 | 441.6 | 468.8 | 455.2 | -27.2 | -15.0 | -13.6 | 9.6 |
| CAR-R3AG-2016-2017 | 400.4 | 407.0 | 406.0 | -6.6 | -5.7 | -1.0 | 4.7 |
| CAR-R3AP-2016-2017 | 365.1 | 394.0 | 380.2 | -28.9 | -17.7 | -13.8 | 12.0 |
| CAR-R3UP-2016-2017 | 313.5 | 345.8 | 330.1 | -32.3 | -19.1 | -15.7 | 11.6 |
| CCR-ABG-2013-2014 | 256.5 | 259.0 | 258.0 | -2.5 | -2.2 | -1.0 | 2.1 |
| CCR-ABG-2014-2015 | 168.6 | 172.7 | 171.7 | -4.1 | -3.7 | -0.9 | 3.5 |
| CCR-ABG-2015-2016 | 171.5 | 174.3 | 174.8 | -2.8 | -1.1 | 0.4 | 3.8 |
| CCR-ABP-2014-2015 | 166.1 | 174.8 | 172.7 | -8.8 | -5.4 | -2.2 | 6.2 |
| CCR-ABP-2015-2016 | 171.1 | 177.1 | 177.0 | -6.0 | -2.1 | -0.1 | 5.9 |
| CCR-R2G-2014-2015 | 105.7 | 106.3 | 108.1 | -0.6 | 8.1 | 1.8 | 3.4 |
| CCR-R2G-2015-2016 | 186.5 | 189.3 | 188.3 | -2.8 | -2.0 | -1.1 | 2.3 |
| CCR-R2G-2013-2014* | 275.5 | 279.6 | 276.5 | -4.0 | 0.4 | -3.1 | 3.0 |
| CCR-R3AG-2013-2014 | 222.9 | 224.1 | 224.6 | -1.2 | -1.0 | 0.4 | 2.5 |
| CCR-R3AG-2014-2015* | 85.8 | 86.8 | 88.1 | -1.1 | -0.4 | 1.3 | 2.6 |
| CCR-R3UG-2013-2014 | 183.4 | 185.2 | 184.4 | -1.9 | -1.3 | -0.8 | 2.5 |
| CCR-R3UG-2014-2015* | 72.3 | 73.9 | 75.6 | -1.6 | -0.7 | 1.7 | 3.0 |
| FMG-R2P-2015-2016 | 1036.7 | 1053.8 | 1042.1 | -17.1 | -13.0 | -11.7 | 3.4 |
| FMG-R3AP-2015-2016* | 828.1 | 849.1 | 832.6 | -21.0 | -15.6 | -16.5 | 2.6 |
| HKL-R2G-2016-2017* | 748.5 | 755.0 | 754.0 | -6.5 | -0.5 | -1.0 | 5.1 |
| HKL-R3AG-2016-2017 | 423.9 | 437.5 | 438.0 | -13.6 | 39.4 | 0.5 | 11.1 |
| HKL-R3AP-2016-2017 | 385.4 | 403.0 | 399.5 | -17.6 | -3.7 | -3.5 | 10.3 |
| HKL-R3UG-2016-2017 | 320.5 | 328.3 | 329.2 | -7.8 | -2.2 | 0.9 | 7.8 |
| SOD-R2P-2016-2017 | 215.0 | 238.4 | 234.7 | -23.4 | -7.4 | -3.7 | 15.7 |
| SOD-R3AP-2016-2017 | 187.7 | 212.9 | 207.8 | -25.2 | -8.9 | -5.1 | 16.7 |
| SOD-R3UP-2016-2017 | 180.9 | 194.1 | 192.0 | -13.2 | -4.1 | -2.2 | 9.4 |
| WFJ-R2P-2016-2017* | 595.4 | 715.1 | 706.6 | -119.7 | -1.5 | -8.5 | 102.4 |





| | | | | | | | |
|---|---|---|---|---|---|---|---|
| **WFJ-R3AP-2016-2017** | 375.4 | 605.7 | 598.0 | -230.3 | 13.2 | -7.7 | 208.6 |
| **WFJ-R3UP-2016-2017\*** | 246.6 | 434.6 | 423.6 | -188.0 | 0.4 | -11.0 | 167.0 |
| **XBK-AP-2013-2014** | 83.8 | 91.9 | 90.7 | -8.1 | -4.2 | -1.2 | 4.9 |
| **XBK-AP-2014-2015** | 49.5 | 59.5 | 58.1 | -10.0 | -6.6 | -1.4 | 7.1 |
| **XBK-AP-2015-2016** | 61.1 | 74.9 | 71.8 | -13.7 | -9.4 | -3.1 | 8.2 |
| **XBK-DAG-2013-2014** | 131.4 | 136.0 | 134.2 | -4.6 | -3.9 | -1.8 | 3.3 |
| **XBK-DAG-2014-2015** | 104.3 | 111.0 | 108.5 | -6.7 | -3.4 | -2.4 | 5.6 |
| **XBK-DAG-2015-2016** | 90.2 | 97.1 | 95.5 | -7.0 | -5.5 | -1.6 | 5.2 |
| **XBK-R2G-2013-2014** | 167.2 | 170.2 | 170.4 | -3.0 | -2.8 | 0.2 | 2.3 |
| **XBK-R2G-2015-2016** | 71.1 | 75.5 | 75.7 | -4.4 | -4.1 | 0.3 | 3.8 |
| **XBK-R2P-2014-2015** | 110.3 | 119.2 | 114.9 | -8.8 | -7.6 | -4.2 | 3.7 |
| **XBK-R2P-2015-2016** | 80.4 | 92.6 | 91.2 | -12.2 | -5.6 | -1.3 | 9.4 |
| **XBK-R3AG-2013-2014** | 97.7 | 100.7 | 101.4 | -3.0 | -2.9 | 0.7 | 3.7 |
| **XBK-R3AG-2014-2015** | 73.0 | 78.3 | 76.9 | -5.3 | -2.8 | -1.4 | 4.7 |
| **XBK-R3AG-2015-2016** | 72.7 | 78.2 | 77.8 | -5.5 | -5.0 | -0.4 | 5.2 |
| **XBK-R3UG-2013-2014** | 83.1 | 89.6 | 90.2 | -6.5 | 3.8 | 0.6 | 7.3 |
| **XBK-R3UG-2014-2015** | 56.4 | 63.8 | 62.3 | -7.5 | -3.0 | -1.6 | 7.2 |
| **XBK-R3UG-2015-2016** | 69.5 | 76.2 | 75.2 | -6.7 | -4.2 | -1.0 | 5.7 |


**Table A2: A description of the different shield/gauge configurations used in table A1.**

| Code | Description |
|---|---|
| *R2* | DFAR Reference (SPICE) |
| *R3* | Alter or Unshielded Reference (SPICE) |
| *A* | Single Alter shield |
| *U* | Unshielded |
| *DA* | Double Alter shield |
| *B* | Bush shield |
| *P* | Pluvio gauge |
| *G* | Geonor gauge |






**Figure 1: NAF-SEG data flowchart**





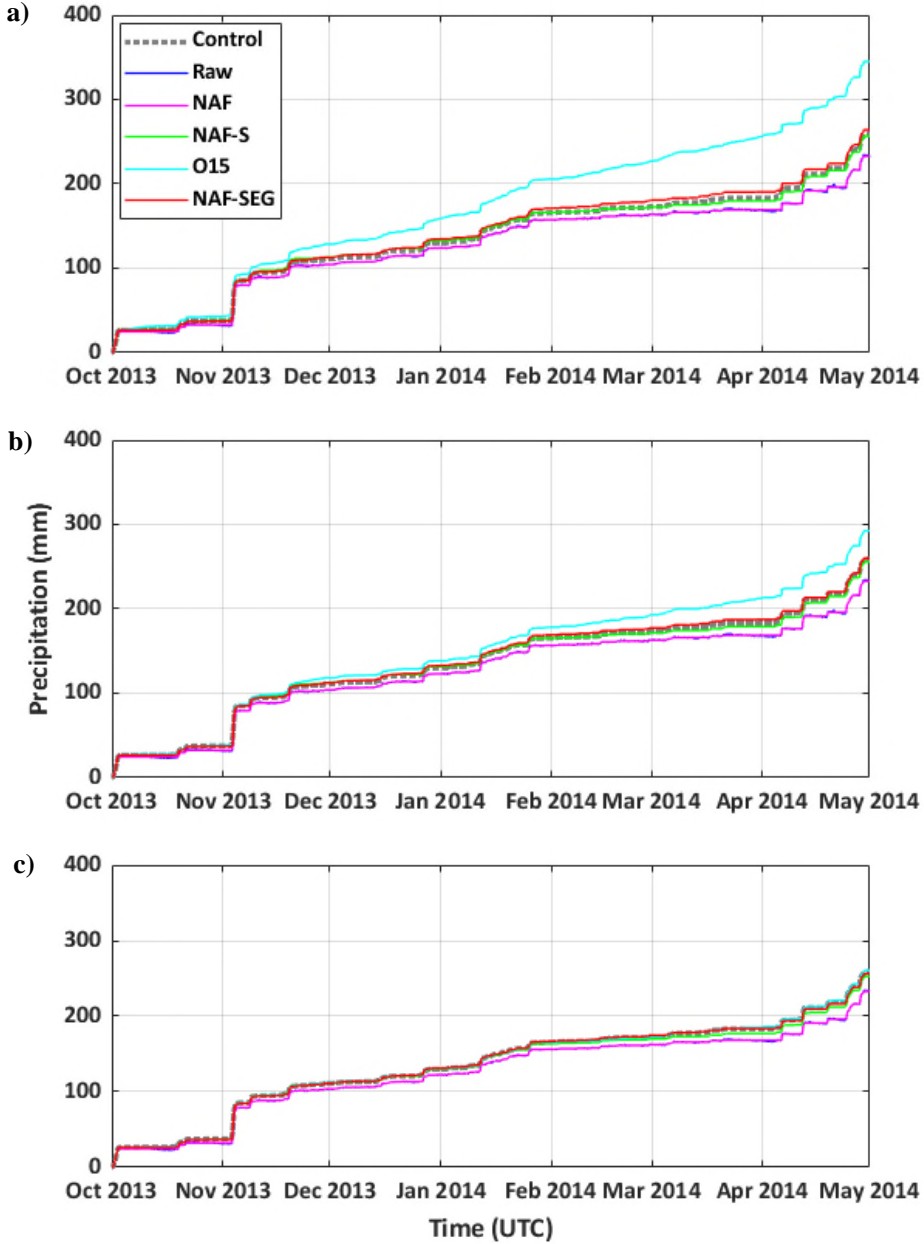

**Figure 2: Time series of simulated precipitation gauge bucket weight with synthetic evaporation and varying levels of synthetic noise and diurnal oscillations (A - high noise; B - med noise; C - low noise).**





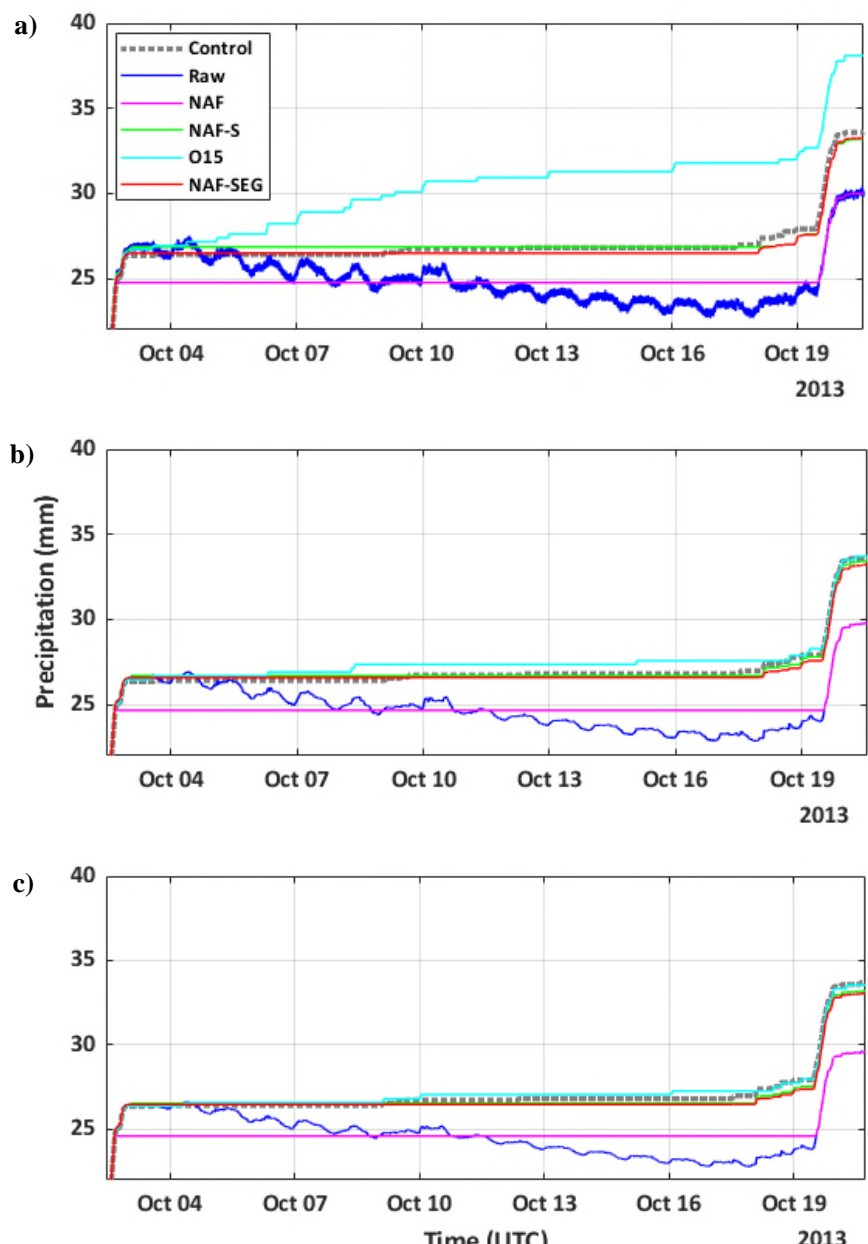

**Figure 3: Time series of simulated precipitation gauge bucket weight (zoomed into the first evaporation event) with synthetic evaporation and varying levels of synthetic noise and diurnal oscillations (A - high noise; B - med noise; C - low noise).**




Figure 4: Box and whisker plots of (a) Pearson r, (b) RMSD and (c) bias in seasonal total precipitation relative to the reference for each of the evaluated filtering techniques (NAF-SEG, NAF, and O15) as compared to the reference technique (NAF-S) for the 44 unprocessed time series.

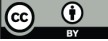



a)

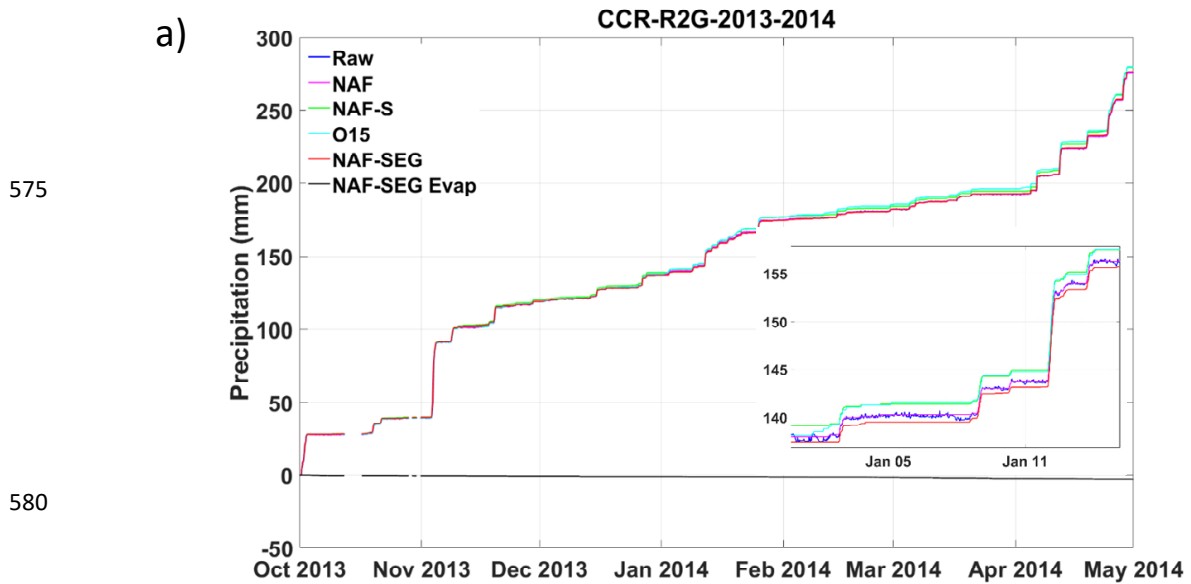



b)



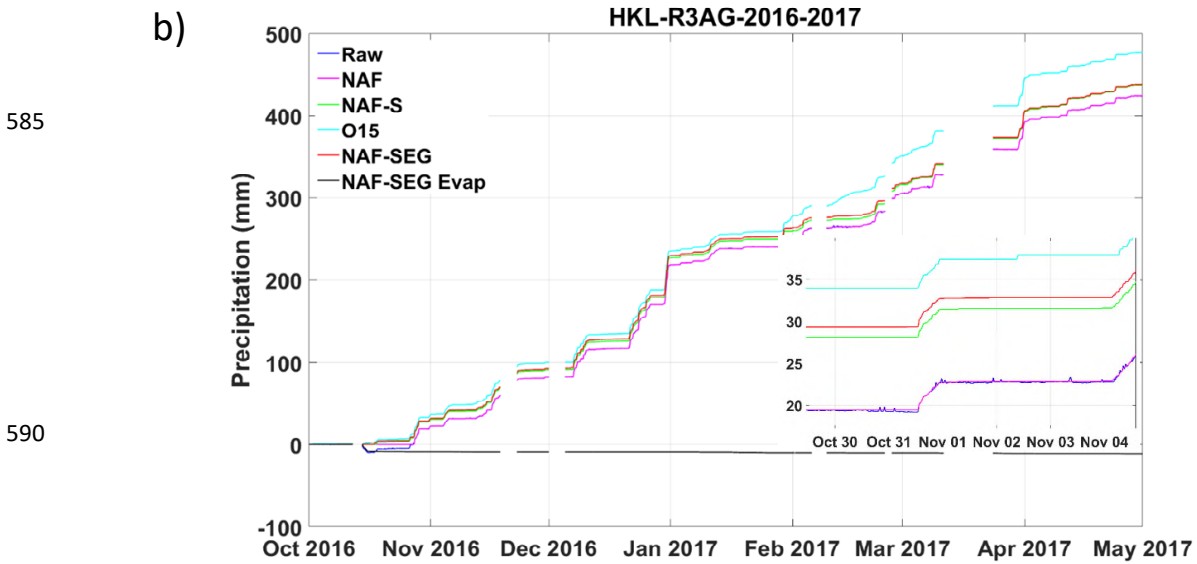




c)

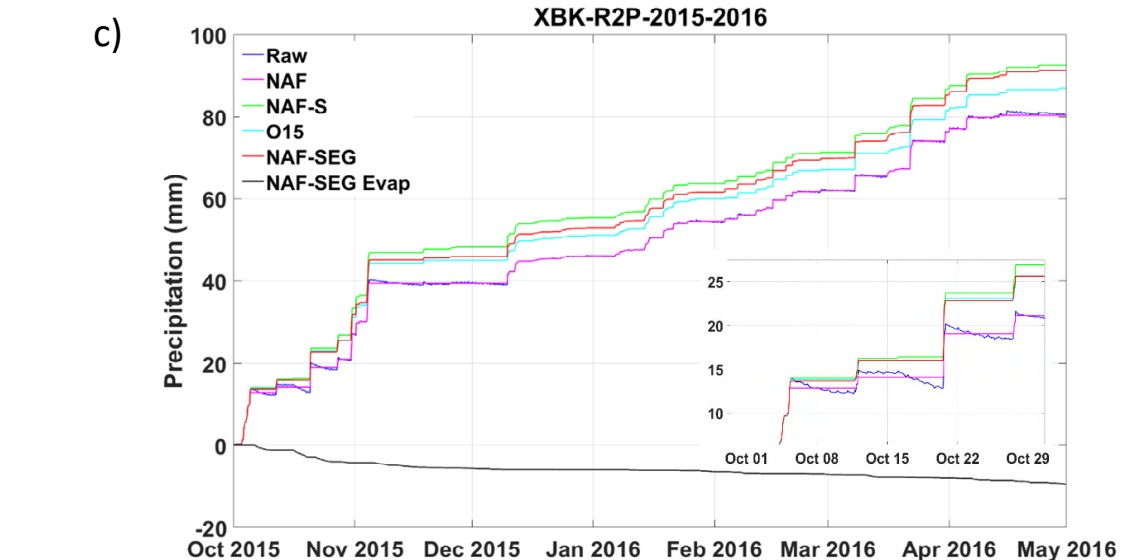



d)

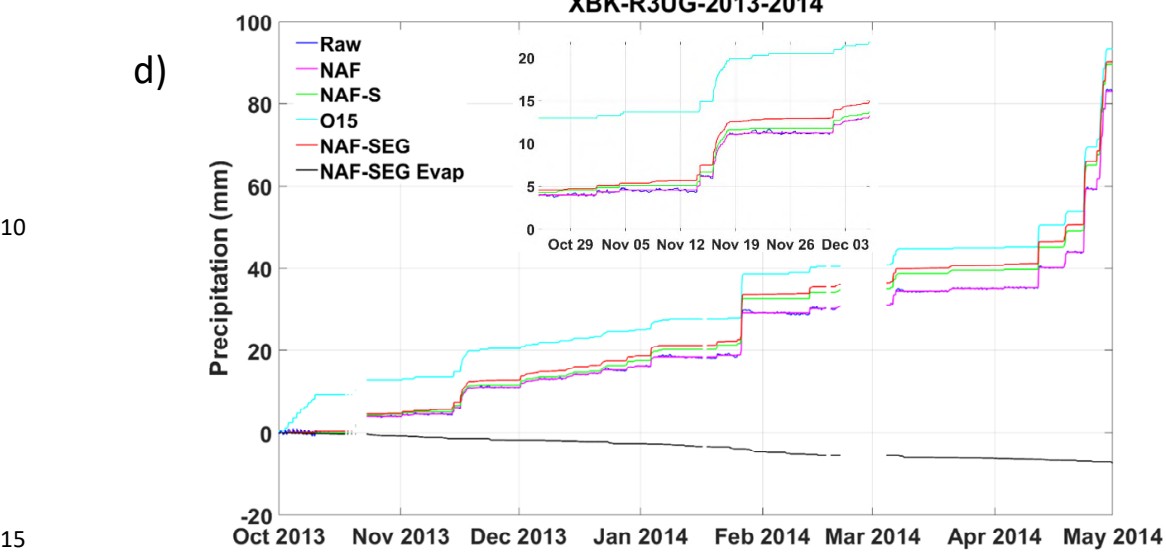



**Figure 5: Time series of observed precipitation gauge bucket weight processing (NAF, NAF-S, O15, and NAF-SEG) along with the NAF-SEG evaporation estimate for a) Caribou Creek R2G 2013-2014, b) Haukeliseter R3AG 2016-2017, c) Bratt's Lake R2P 2015-2016, and d) Bratt's Lake R3UG 2013-2014. Insets show a zoomed example with consistent vertical scaling to illustrate the issues and filter performance relative to each time series.**
