# Peer review of "An improved post-processing technique for automatic precipitation gauge time series"

_Atmospheric Measurement Techniques, 2019_

## Referee Comment (RC1) · Anonymous Referee #1 · 14 Jan 2020

Post-processing is essential to the automated accumulating precipitation gauges, although it is just a filtering technique. This study proposed an improved post-processing technique to tackle the noise caused by diurnal oscillations and drift from the evaporation of the bucket contents. Comparing with other techniques, the major advantage of the suggested one is its fully-automated processing with a 24-hour latency. Generally, this study is well written and presented. I am happy to see this paper published in the Atmospheric Measurement Techniques. But the following issues should be addressed properly before the paper can be considered for publication.

1. For users, people would like to know what are the performances of the filter for all-weather precipitation. Compared to a much smaller amount of solid precipitation in the cold season, testing the filter might be more important in the warm season. First of all,

the drift from evaporation in the warm season can be much more serious in most cold regions, and the evaporation rate can be much larger. Secondly, the noise features can be quite different between warm and cold seasons. Thus, to make the conclusion more solid for both rainfall and solid precipitation, I would like to see the performance for the warm season.

2. Compared to the robustness of NAF-S, the validity of NAF-SEG is closely related to the setting of the minimum threshold P* = 0.001 mm. Although the authors assert it is somewhat arbitrary within the tested conditions of solid precipitation measurements, it might be challenging for the noise features in the warm season. Considering the more variability of precipitation and stronger evaporation in the warm season, further exploration in the point is necessary. In addition, there is no validation for the raw precipitation data when using the filters. Therefore, validation using independent measurements from the tipping bucket would be very helpful for the filtered measurements from the accumulating gauges.

3. As we know the performances of the filters are slightly related to the climate of the observed sites. Further discussion of the relationships between the biases for the 44 raw time series would help understand the validity of the filters in different environments.

Mineral comments: (1) P1-L5: If my understanding is correction, this study is talking about the weight-based precipitation gauge. It is quite confusing when using 'automatic precipitation gauge', 'automated accumulating precipitation gauge' and 'automated accumulating (weighting) precipitation gauge'. (2) P12-L406: 'his' to 'this'.

---

## Referee Comment (RC2) · Anonymous Referee #2 · 17 Jan 2020

This paper presents the evaluation of four post-processing algorithms for the filtering of precipitation time series measured by weighing gauges. One of those algorithms is applied operational for weighing gauges within the Environment Canada network and is implemented directly at the data logger, working near real time, while the three others requires a 24 hour segment of the time series and are thus processed after data are transferred from the sites to Environment Canada. The study compares the performance of all four algorithms applying both a synthetic dataset and a real-world dataset with data from international sites in different climates and configurations. Noise on weighing gauge signals is a common and well-known problem. Mechanical vibrations and electro-magnetic disturbances cause high frequent noise, while evaporation, temperature changes and leakages may cause more low frequent noise, oscillations

or drift on the signal. Given the various sources, the exact structure of the noise of a single gauge at any given site may vary a lot. Filtering algorithms are applied worldwide, but documentation of these are sparse. Weather services with weighing gauges in their network may have developed their own filters, often only described in an internal note while instrument providers may have added an inbuilt and proprietary filtering algorithm into the instrument electronics, prone to firmware updates with unknown changes. This situation makes it difficult to analyze precipitation measurements across networks because detailed information on the post-processing procedures are missing. Therefore, this paper is very valuable and falls absolutely inside the scope of the Journal "Atmospheric Measurement Techniques". I specially want to thank the authors for the good description of their algorithm and the announcement to make the code available on an online repository. This will open for further exchange and comparison of existing filter algorithms, thus positively affecting any across-network comparisons and cooperation and hopefully encourages other precipitation-gauge network holders and instrument providers to describe and share their post processing methods. The paper is well written and readable. However, at some points, some more detailed or further clarified information would help the overall understandability of the paper. I recommend accepting the paper, requiring minor revisions. Attached, a list of more specific comments.

Please also note the supplement to this comment:
https://www.atmos-meas-tech-discuss.net/amt-2019-423/amt-2019-423-RC2-supplement.pdf
* * *
[Figure]

**Supplement:**

Review of «**An improved post-processing technique for automatic precipitation gauge time series**» by Amber Ross, Craig D. Smith, and Alan Barr, *Manuscript Number: amt-2019-423.*

**List of specific comments**

Throughout the paper

In the introduction, you do name three post-processing challenges: mechanical and electrical interference, diurnal oscillations, and evaporation of the bucket contents. While you treat mechanical and electrical interferences and evaporation explicit with your filtering method, possible diurnal variations as the temperature dependency of the measurement device are treated more implicit with the introduction of a 24h measurement window, where you can assume similar temperatures at the end and beginning of the cycle without explaining this part of the algorithm.

However, synoptic changes can make this assumption not valid and thus some of the detected precipitation or evaporation can be due to temperature changes independent of a diurnal cycle. I don't think that this is problematic for your results, but I suggest to discuss this issue throughout your paper .

Section 2.4 Segmented Neutral Aggregating Filter, page 6, algorithm description and Figure 1 in Appendix (flowchart):

Please clarify that the measurement interval in your case is actually a minute or has to be a much shorter interval than the 24 hour segments. I think it may also help if you punctuate (in addition to the use of indices) when you are treating the 24 hour segment as one: i.e. all individual measurements from one interval is assigned the same flag "precipitating", "evaporating" or "neither E nor P", and when you are treating minute by minute (each single measurement can get its individual $P(i)$ or $E(i)$, and from step 4 you evaluating minute values).

Please reword 6c. While the flowchart indicates clearly that if answers on both questions 6a and 6b are no, precipitation and evaporation are set to zero. That includes also (and especially) those cases where not all three overlapping windows do agree.  After repeated reading of sentence "6c" and a look on the flowchart, I actually understood that 6c could be understand this way. However, I do suggest to rewrite and clarify the point that you are also looking for disagreement between the three flags here and not only for those cases where all three flags indicate (in agreement) no precipitation nor evaporation (the latter is how I understood the sentence when reading it the first time).

Section 3.1. Testing with pre-procesed (control) precipitation data

The description of the creation of the synthetic signal is very informative. I was searching after a figure illustrating the level of noise visual. Maybe you can hint other readers that figure 3 in the appendix actually have that kind of visualization. Also, in contrast to figure 2, where the difference between the three noise levels is not visible due to the higher overall changes.

Section 3.2 Testing with raw precipitation data

From your description, it becomes clear, that you actually include a double filtering. The QC process used for the WMO-SPICE analysis is already cleaning and smoothing the data series before you apply the described filters of this study.  That is off course no problem, but I do think it is important enough that it should be mentioned earlier in the paper and also be taken up again in the discussion of the results. I am wondering especially about:

- Have you tried to apply your filtering algorithm without this additional SPICE-filtering and QC?
- In the operational use of your O15 filter, you calculate a 5-minute mean prior to filtering. Do you assume that the 5-minute average calculation would do about the same as the 4-minute Gaussian filter of the SPICE-algorithm?
- In case of your study, did you still use the 5-minute averaging step of the O15 filter after applying the 4 minute Gaussian filter of the SPICE-algorithm?
- Do you think a quality control of the time series is necessary before applying the filter? Especially when thinking of the O15 filter, but also for the other filters, it may be more usual to apply the complete or parts of the quality control on the filtered (with your algorithm) data - what are the advantages/disadvantages of either way?

Chapter 5 – Discussion, lines 364-372:

I think the necessity of antifreeze and oil, also when an algorithm is applied, is valuable information, which should also occur in the conclusions (in a slightly shorter form)

Use of Ott Pluvio2 data – throughout the paper

In the Introduction (lines 68-71), you describe that you are using somewhat processed values of the Ott Pluvio2 gauges. In Section 3.2 (lines 285ff.), however, you do not distinguish between data from Geonor or Pluvio2.

Do you apply the SPICE-algorithm on the preprocessed or the raw bucket data from Ott Pluvio2? To my understanding, the SPICE algorithm is meant to be applied to the raw bucket data. Depending on what you actually did, Pluvio2 and Geonor data may have been treated significantly different and I wonder if that should be visible in your results. Do you see any effect of a possible different treatment of the data from the different gauges? I was surprised to see that evaporation was detected in a similar manner for Pluvio2 gauges as for Geonor gauges even if (after my understanding) evaporation for Pluvio2 gauges were already treated from the inbuilt algorithm and thus probably be "treated" twice.

Appendix

Flowchart and figures 2 and 3 contain relevant information and I suggest moving them from the appendix into the main text.

Appendix, Figure 5

The lines are very thin and difficult to distinguish; the evaporation line seems to be almost constantly zero, due to the different orders of magnitude. You try to overcome some of these issues with the smaller inserts, but those makes the plots "untidy" and difficult to understand. Please consider to use several plots, shorter time intervals, or some other way to improve the quality of these figures.

---

## Author Comment (AC2) · 21 Apr 2020

Thank you very much for the review of this manuscript. We understand that this takes a significant commitment and we very much appreciate the time and the effort spent on the review to help us improve this manuscript. Our responses are attached as a supplement.
* * *

---

## Author Response (AR1)

**Author's response to interactive comments by Referee #1**

Post-processing is essential to the automated accumulating precipitation gauges, although it is just a filtering technique. This study proposed an improved post-processing technique to tackle the noise caused by diurnal oscillations and drift from the evaporation of the bucket contents. Comparing with other techniques, the major advantage of the suggested one is its fully automated processing with a 24-hour latency.

Generally, this study is well written and presented. I am happy to see this paper published in the Atmospheric Measurement Techniques. But the following issues should be addressed properly before the paper can be considered for publication.

1. For users, people would like to know what are the performances of the filter for all-weather precipitation. Compared to a much smaller amount of solid precipitation in the cold season, testing the filter might be more important in the warm season. First of all, the drift from evaporation in the warm season can be much more serious in most cold regions, and the evaporation rate can be much larger. Secondly, the noise features can be quite different between warm and cold seasons. Thus, to make the conclusion more solid for both rainfall and solid precipitation, I would like to see the performance for the warm season.

AC: The reviewer makes a good point. It is important that users know the performance of the filter for each season. For this analysis, however, we chose winter for several reasons:

1)The data set that we had available to us was a mainly cold season precipitation data set originating from the WMO-SPICE (or post-SPICE) project. We chose this data set because it had a known quality. This data came with relatively meticulous metadata such as service logs and field notes so that we were confident in our ability to quality control this data to allow for a level playing field for each filter. The quantity and quality of the warm season data from SPICE is reduced and much of this wasn't readily available at the time that this analysis was undertaken.

2)The signal to noise ratio is always lower in the winter due to lower (generally) precipitation rates. In our opinion, this makes filtering of winter data substantially more difficult than filtering of summer data, where small absolute errors are less likely to be large relative errors.

3)We know that the evaporation signal in the SPICE data is significant during the shoulder seasons (Fig. 5c as an example), perhaps even more significant relative to total precipitation than evaporation during the summer months. We felt that this cold (shoulder) season evaporation would be a considerable challenge to the filters.

To help answer the question about warm season performance, we ran a separate control experiment on the warm season and added 11 available unprocessed warm season time series to the analysis.

The results of the warm season control analysis suggested that in general, the performance of NAF remained consistent, NAF-S improved, O15 became even more unstable, and the performance of NAF-SEG dropped somewhat, apparently because of a lower recovery rate for evaporation (Table 4). However, NAF-SEG continued to outperform NAF and O15 in nearly every metric.

Although we don't have nearly as much summer data as winter data due to the focus of SPICE, we filtered 11 available warm season time series of known data quality for comparison with winter results. All filters showed a slight reduction in performance for the warm season with an increase in RMSD vs. the cold season (Table 5). The biggest increase in RMSD was in the O15 filter (increase of nearly 0.03 mm).

Action: We have clarified our justification in the methods section for focussing on the filtering of cold season data but since we agree that assessing the performance during the warm season is also important, we have added the warm season control exercise and the addition of the 11 warm season time series to the methods sections and summarized the results from these experiments in the appropriate sections, including an update of Tables 2-5 and the addition of Table A2. A substantial addition to the Discussion section was made to discuss the results of both the pre-processed and unprocessed warm-season testing.

2. Compared to the robustness of NAF-S, the validity of NAF-SEG is closely related to the setting of the minimum threshold P* = 0.001 mm. Although the authors assert it is somewhat arbitrary within the tested conditions of solid precipitation measurements, it might be challenging for the noise features in the warm season. Considering the more variability of precipitation and stronger evaporation in the warm season, further exploration in the point is necessary. In addition, there is no validation for the raw precipitation data when using the filters. Therefore, validation using independent measurements from the tipping bucket would be very helpful for the filtered measurements from the accumulating gauges.

AC: Analysis not shown in the manuscript tests the sensitivity of NAF-SEG to P* and found little to no sensitivity (which was actually somewhat surprising), but the reviewer would be correct in assuming that the tests were only performed on winter data. This can be tested relatively easily using the same data used to address comment 1. As for using tipping bucket rain gauge (TBRG) data as a reference (during warm season tests), the authors feel that TBRG data has it's own inherent problems and would be not be conducive for use as a reference or even as an independent validation due to known

issues with splash, siphoning delays, unknown maintenance issues, calibration, etc. We think that the greatest potential for future improvements could be the incorporation of present weather detectors or disdrometers into the filtering process for identifying light and false events.

Action: Using the same observed warm season time series data discussed above, the NAF-SEG filter was tested using different P* values ranging from 0.0001 mm to 0.5 mm. This is a very similar test to that run using cold season data. Results were similar to the cold season in that the response in the metrics was subtle up to 0.05, only dropping off substantially at 0.5. This was added to the discussion section. Also, in response to the reviewers question about independent validation, we added a paragraph to the discussion section suggesting that the use of present weather detectors or optical disdrometers could be explored to validate or improve filtering techniques. The use of a TBRG or other precipitation detectors is out of the scope of this current analysis.

3. As we know the performances of the filters are slightly related to the climate of the observed sites. Further discussion of the relationships between the biases for the 44 raw time series would help understand the validity of the filters in different environments.

AC: The reviewer makes a good point. However, it's not the filter that is impacted by climate but rather the behaviour of the precipitation gauge. Although climate is a factor (e.g. wind vibration, temperature signals, evaporation), there are many non-climate related factors that also have a significant impact on gauge performance, such as electrical interference, service interval, or even the actual installation of the gauge and infrastructure. These are very difficult to isolate from the impacts of climate. Examining the impact of these factors, including climate conditions, on the signal behaviour of the gauge was discussed in the SPICE final report and was recommended for further analysis, but understanding these impacts are out of scope for this analysis.

Action: In the discussion section, we suggest that filtering techniques (whether this filter or others) could be improved by better understanding the cause of signal noise and filtering made easier by reducing signal noise during measurement.

Minor comments:

(1) P1-L5: If my understanding is correction, this study is talking about the weight-based precipitation gauge. It is quite confusing when using 'automatic precipitation gauge', 'automated accumulating precipitation gauge' and 'automated accumulating (weighting) precipitation gauge'.

AC: These gauges are in fact accumulating automated weighing precipitation gauges.

Action: We will define this better and make the nomenclature consistent to "automated weighing gauge"

(2) P12-L406: 'his' to 'this'

Action: done

**Author's response to interactive comments by Referee #2**

Throughout the paper

In the introduction, you do name three post-processing challenges: mechanical and electrical interference, diurnal oscillations, and evaporation of the bucket contents. While you treat mechanical and electrical interferences and evaporation explicit with your filtering method, possible diurnal variations as the temperature dependency of the measurement device are treated more implicit with the introduction of a 24h measurement window, where you can assume similar temperatures at the end and beginning of the cycle without explaining this part of the algorithm. However, synoptic changes can make this assumption not valid and thus some of the detected precipitation or evaporation can be due to temperature changes independent of a diurnal cycle. I don't think that this is problematic for your results, but I suggest to discuss this issue throughout your paper.

AC: True, a non-diurnal temperature fluctuation may have an impact on signal noise and therefore impact the NAF-SEG filtering. It might be worthwhile looking at strong non-diurnal gradients to determine if that could explain when and why the filters have a reduced performance.

Action: The following sentences were added to the discussion: "There may also be a decrease in the performance of NAF-SEG when signal noise is due to non-cyclical temperature fluctuations, such as those that occur during strong synoptic events. Although this wasn't explicitly assessed, it may be a situation that a user should be aware of."

Section 2.4 Segmented Neutral Aggregating Filter, page 6, algorithm description and Figure 1 in Appendix (flowchart): Please clarify that the measurement interval in your case is actually a minute or has to be a much shorter interval than the 24 hour segments. I think it may also help if you punctuate (in addition to the use of indices) when you are treating the 24 hour segment as one: i.e. all individual measurements from one interval is assigned the same flag "precipitating", "evaporating" or "neither E nor P", and when you are treating minute by minute (each single measurement can get its individual $P(i)$ or $E(i)$, and from step 4 you evaluating minute values).

Action: added the following paragraph to Section 2.4:

"The measurement interval used in this analysis to evaluate NAF, NAF-S, and NAF-SEG is 1-min. This interval is used here because it was chosen as the preferred interval for archiving of the SPICE data, and was therefore available for this analysis. NAF has been shown to work on data of larger intervals (i.e. 30 min in Pan et al.,2016) and there is no reason why NAF-SEG could not be used with larger intervals as well, provided that the intervals are considerably shorter than the 24-hour window (i.e. 30 minutes or less)."

Please reword 6c. While the flowchart indicates clearly that if answers on both questions 6a and 6b are no, precipitation and evaporation are set to zero. That includes also (and especially) those cases where not all three overlapping windows do agree. After repeated reading of sentence "6c" and a look on the flowchart, I actually understood that 6c could be understand this way. However, I do suggest to rewrite and clarify the point that you are also looking for disagreement between the three flags here and not only for those cases where all three flags indicate (in agreement) no precipitation nor evaporation (the latter is how I understood the sentence when reading it the first time).

AC: Agreed

Action: The sentence now reads "a.  For intervals which are not precipitating (a) or evaporating (b), i.e. when the flag from all three overlapping windows indicates the absence of both precipitation and evaporation, or when the three flags do not agree with each other, P(i) and E(i) are set to zero.

Section 3.1. Testing with pre-procesed (control) precipitation data

The description of the creation of the synthetic signal is very informative. I was searching after a figure illustrating the level of noise visual. Maybe you can hint other readers that figure 3 in the appendix actually have that kind of visualization. Also, in contrast to figure 2, where the difference between the three noise levels is not visible due to the higher overall changes.

Action: we indicated near the end of Section 3.1 that the various noise levels in the synthetic data can be visualized in Figure 3.

Section 3.2 Testing with raw precipitation data

From your description, it becomes clear, that you actually include a double filtering. The QC process used for the WMO-SPICE analysis is already cleaning and smoothing the data series before you apply the described filters of this study. That is off course no problem, but I do think it is important enough that it should be mentioned earlier in the paper and also be taken up again in the discussion of the results.

AC: agreed. This is already noted in Section 3.2

Action: A brief discussion of the impacts of the Gaussian filter is included in the discussion section.

I am wondering especially about:

Have you tried to apply your filtering algorithm without this additional SPICE-filtering and QC?

AC: We have in more recent work but not in this work. Preliminary results suggest that the Gaussian filter has little impact on the performance of NAF-SEG but it may help O15. Unfortunately, it was built into the SPICE data QC, some of which was completed before we received the data, so it was maintained for consistency.

In the operational use of your O15 filter, you calculate a 5-minute mean prior to filtering. Do you assume that the 5-minute average calculation would do about the same as the 4-minute Gaussian filter of the SPICE-algorithm?

AC: I think that the 5-minute mean would be more aggressive than the Gaussian filter. This is considered part of the O15 filter, not a pre-filter process. This is what is implemented on the operational data loggers.

In case of your study, did you still use the 5-minute averaging step of the O15 filter after

applying the 4 minute Gaussian filter of the SPICE-algorithm?

AC: Yes, the 5-min mean was applied after the Gaussian smoothing, which probably gave the O15 filter more of an advantage with this test data than it would get in real-time on the data logger. This should be pointed out in the discussion.

Action: added the following sentences to the discussion section: "It should also be noted that the unprocessed data in these tests were pre-filtered with a Gaussian filter with a 4-min window, which was integrated into the SPICE quality control process, prior to testing the algorithms. This likely resulted in the O15 filter performing better than it would have in the operational setting, but this was not confirmed."

Do you think a quality control of the time series is necessary before applying the filter?

AC: Yes, some quality control of the time series is necessary before applying the filter. We can speculate that the Gaussian filter has a negligible impact on performance, but artifacts such as those caused by gauge servicing (as an example) need to be removed. Generally, these don't appear in the operational data due to servicing protocols (i.e. the data is suspended during servicing) but simple range checks and jump filters would prevent other artifacts from impacting the data.

Action: we added the recommendation for a post-processing enhanced quality control procedure, along with the archiving and re-processing of the 1-min operational data in non-real-time, to the final sentence in the conclusions section.

Especially when thinking of the O15 filter, but also for the other filters, it may be more usual to apply the complete or parts of the quality control on the filtered (with your algorithm) data - what are the advantages/disadvantages of either way?

AC: The advantage, and perhaps the only advantage, of the O15 filter is that it can be deployed operationally (i.e. on the data logger) to work in real-time. This means that only rudimentary quality control measures are available. The O15 filter definitely benefits from pre-filtering data quality control, as do the other filters. Testing the impacts of QC procedures was out of scope for this analysis (which is why we used the pre-tested SPICE process), but we can certainly see the benefit of testing and implementing enhanced QC processes in operational data management systems as a first step to an integrated precipitation post-processing technique. I don't see any disadvantages, other than keeping processes separate may make revisions, documentation, and implementation less complex.

Action: see above

Chapter 5 – Discussion, lines 364-372:

I think the necessity of antifreeze and oil, also when an algorithm is applied, is valuable information, which should also occur in the conclusions (in a slightly shorter form)

AC: agreed

Action: added "This, in combination with routine site servicing to pre-empt evaporation and other sources of noise, can result in improved operational precipitation data." As the final statement in the conclusions section.

Use of Ott Pluvio2 data – throughout the paper

In the Introduction (lines 68-71), you describe that you are using somewhat processed values of the Ott Pluvio2 gauges. In Section 3.2 (lines 285ff.), however, you do not distinguish between data from Geonor or Pluvio2. Do you apply the SPICE-algorithm on the preprocessed or the raw bucket data from Ott Pluvio2? To my understanding, the SPICE algorithm is meant to be applied to the raw bucket data. Depending on what you actually did, Pluvio2 and Geonor data may have been treated significantly different and I wonder if that should be visible in your results. Do you see any effect of a possible different treatment of the data from the different gauges? I was surprised to see that evaporation was detected in a similar manner for Pluvio2 gauges as for Geonor gauges even if (after my understanding) evaporation for Pluvio2 gauges were already treated from the inbuilt algorithm and thus probably be "treated" twice.

AC: The description in the introduction describes what is available as output from the Pluvio2, but states that some users (including the authors) would like to bypass "further processing" and have the option to "complete their own post-processing of the data in its rawest form." The output used for testing from the Pluvio2 was the real-time bucket weight output. This data, according to OTT, is pre-processed by the onboard algorithm in that it is a high frequency measurement which is internally corrected for temperature and vibration effects on the load cell. This output is NOT corrected for evaporation. This is the "rawest" output product available from this gauge and is as close as possible to the raw unprocessed data derived from the Geonor. The SPICE algorithm was designed to be applied to both the raw Geonor ouput and the Pluvio2 real-time bucket weight data in the same manner, with only parameter changes for range and jump thresholds. Having said that, the signal noise for the gauges have both similarities and differences, but are treated the same by all four filtering algorithms under test.

Action: clarified the output source of both the Geonor and Pluvio2 in paragraph 2 of Section 3.2

Appendix

Flowchart and figures 2 and 3 contain relevant information and I suggest moving them from the appendix into the main text.

AC: We apologize as this was a mistake in the order of the tables and figures. Only Table A1 is meant to be in the Appendix while the figures, including the flow chart, should be integrated into the manuscript during final publication.

Appendix, Figure 5

The lines are very thin and difficult to distinguish; the evaporation line seems to be almost constantly zero, due to the different orders of magnitude. You try to overcome some of these issues with the smaller inserts, but those makes the plots "untidy" and difficult to understand. Please consider to use several plots, shorter time intervals, or some other way to improve the quality of these figures.

AC: agreed. These will be improved when we re-submit the manuscript for final publication.

[revised manuscript text omitted]
. Although RMSD increased for each of the filters for the warm-season, the increase in O15 was the largest, and consistent with the pre-processed control experiments.Fewer warm-season test cases were examined, largely due to data availability, since the unprocessed data used in this analysis were obtained from the SPICE project that had a focus on cold-season precipitation intercomparisons. It was expected that warm-

455 ~~season bucket evaporation would be higher relative to the cold-season and that the warm-season noise could exhibit different characteristics and therefore have a negative impact on the performance of the filters. Of the 11 warm-season test cases, O15 only outperformed NAF-SEG in three of those cases (all from XBK). The reason for these three cases cannot be easily explained. Although all of the NAF-SEG warm-season biases were negative (under-estimating), and on average slightly higher than for the cold-season (1.7% and 1.0% for the warm- and cold-season~~

460 Given the smaller warm-season sample size, it was not possible to  determine if the performance differences are significant.  
[revised manuscript text omitted]